# Geospatial interpolation and hydro-geochemical characterization of alluvial aquifers in the Thal Desert, Punjab, Pakistan

Irfan Raza[1], Perveiz Khalid[1], Muhammad Irfan Ehsan[1]*, Qazi Adnan Ahmad[2]*, Shahzada Khurram[1], Rabia Zainab[1], Salman Farooq[1]

1 Institute of Geology, University of the Punjab, Lahore, Pakistan, 2 College of Energy and Mining Engineering, Shandong University of Science and Technology, Qingdao, China

* irfan.geo@pu.edu.pk (MIE); qaa.geo@gmail.com (QAA)

**Data Availability Statement:** All data is in attached file "Supplementary".

## Abstract

This study seeks to assess the hydrogeochemical characteristics of groundwater in the southern part of Thal Desert of Pakistan. The primary focus lies in identifying potential sources of contamination and evaluating their impact on groundwater and the ecosystem. Groundwater samples were collected from diverse sources including shallow hand pumps, tubewells, and dug wells, with depths ranging from 11 to 28 m. A comprehensive analysis was performed to scrutinize the physical, chemical, and microbial attributes of the samples. Utilizing visual aids like the Piper, Durov, and Gibbs diagrams, as well as Pearson correlation, scatter plots, Schoeller diagrams, and pie charts, the study evaluated the groundwater quality and its suitability for consumption. Results indicate that mineral infiltration from rainfall, domestic waste, and industrial effluents significantly affects groundwater quality, leading to widespread salinity. Weathering processes and ion exchange were identified as key factors contributing to elevate levels of bicarbonates, sodium, magnesium, and chloride ions. Employing the Water Quality Index (WQI) on 40 groundwater samples, findings reveal that 52.5% of samples demonstrated poor to not suitable quality, with 27.5% categorized as poor, 2.5% as very poor, and 22.5% not suitable consumption. Conversely, 47.5% of samples showcased good to excellent quality, with 25% rated as good and 22.5% as excellent. These findings provide valuable insights for hydrogeologists to develop appropriate strategies for water treatment and address any concerns related to groundwater quality.

## Introduction

Groundwater stands as a fundamental element in our existence, serving as a primary source for drinking, irrigation, and industrial applications. An estimated one-third of the global population relies on groundwater for household needs, including drinking [1–3]. Groundwater resources play an important impact in economic growth and human health optimization [4, 5]. In comparison to surface water, groundwater is often less susceptible for contamination. However, rehabilitating the contaminated resource, once the groundwater is polluted, is

**Funding:** The authors acknowledge partial funding support from the National Natural Science Foundation of China (Grant No. 42250410333). This funding was received by Dr. Qazi Adnan Ahmed. The funders did play a role in the data collection of this study. They funded the field survey, providing essential equipment such as TDS meters and high-density polyethylene (HDPE) sampling bottles. Additionally, they covered various other field costs, enabling comprehensive and accurate data collection.

**Competing interests:** The authors have declared that no competing interests exist.

extremely challenging [6–9]. Geo-environmental hazards may arise as a consequence of anthropogenic activities and prolonged groundwater exploitation [10, 11]. In order to promote sustainable economic growth in any region, it is essential to advance a hydrogeological dissertation [12, 13].

The quality of groundwater resources has been confirmed to be impacted by both anthropogenic activities like solid waste disposal, unmanaged drainage, unplanned industrial and agricultural activities, urbanization, mining, frequent use of insecticides, pesticides, and agronomic practices as well as natural processes like the quality of recharge waters, mineralogical precipitation, groundwater velocity, interaction with subsurface water sources, and dissolution [3, 14, 15]. Groundwater consistently holds varying concentrations of soluble salts, posing potential health risks to humans. Moreover, these elements can compromise the suitability of groundwater for agricultural activities. Meanwhile, the direct transit, infiltration recharging from surface water, and inter-aquifer interaction of harmful heavy metals can all lead to groundwater contamination [15–18]. Some elements, including Pb, Fe, As, and Cr, have reactivity with inorganic anions, particularly nitrate, sulfate, chlorine, and bicarbonate, and generate species, even though most elements in water have a tendency to form hydrolyzed species [12, 14, 19]. As a result, using low-quality groundwater for drinking or irrigation may have negative impacts on human health or decrease crop yield [15, 20]. Studies have indicated that groundwater pollution contributes significantly to public health challenges, causing 80% of human diseases and 30% of infant mortality in developing nations [21, 22]. The undeniable link between groundwater quality and human well-being emphasizes the vital role of ensuring high water quality standards. Therefore, the evaluation of groundwater quality for drinking and irrigation purposes becomes imperative to safeguard human health and the well-being of all living organisms. This underscores the necessity for ongoing assessments and vigilance to preserve the quality of groundwater resources amid the diverse challenges posed by human activities and industrial processes.

The most precise and effective ways to describe and comprehend the geographical distribution of groundwater are through the use of geospatial tools and earth observation data [23, 24]. Geographic information systems (GIS) have been widely utilized as a tool for evaluating the quality of groundwater due to its dependability, suitability to diverse situations, and capacity to forecast and analyze complex data sets [25, 26]. The effectiveness of GIS also lies in its capability to examine hydrogeochemical and hydrogeological data and give spatially distributed data, which is necessary for well-informed decision-making as well as policy direction [25, 27]. Advanced geostatistical tools, such as Kriging, have been effectively used in recent research to map the spatial fluctuations of groundwater contaminants using point observation [28, 29]. The primary benefit of using geostatistical approaches is that they provide interpolated spatial and temporal variability and give an estimate of the level of uncertainty for each place [29, 30]. Furthermore, Grapher and AquaChem software are employed to graphically depict hydro chemical data on various water quality diagrams, facilitating a comprehensive understanding of the geochemical dynamics within the study area. Researchers have created various WQI models using the weighted arithmetic method to assess water quality. The WQI, a dimensionless number from 0 to 100, categorizes water quality as excellent, good, poor, very poor, or not suitable. It reflects water quality at specific locations and times, based on multiple parameters like pH, TDS, EC, turbidity, and other contaminants. The WQI simplifies complex water quality data into an easily understandable format, making it accessible to non-experts. It is crucial for comparing groundwater quality, managing resources, and selecting cost-effective treatment processes. By capturing the composite impact of various parameters, the WQI effectively communicates water quality information to the public and policymakers. It aids in monitoring trends, assessing policy effectiveness, and prioritizing remediation efforts. Additionally, the

straightforward representation of water quality fosters public awareness and community involvement in water conservation efforts, enhancing environmental monitoring and decision-making.

Numerous research on the quality of groundwater for irrigation and drinking have been conducted in various territories of Pakistan [3, 5, 31, 32]. The study area is predominantly a densely populated area with many buildings and homes. Groundwater is the primary and major source of water for all purposes. The baseline investigation of the geochemical analysis of groundwater resources and their appropriateness for drinking purposes in the study region is demarked by this research. The current study attempts to analyze several geochemical processes in the groundwater in order to better understand the hydrogeochemical framework of the Thal Dessert. The primary objective of the research is: 1) to conduct hydro-chemical characterization of the research area 2) to determine the geogenic and anthropogenic causes of aquifer contamination 3) to evaluate the concentrations and the spatial variations of the major ions ($HCO_3^-$, $Cl^-$, $Mg^{2+}$, $K^+$, $Na^+$, $NO_3^-$, $F^-$, and $SO_4^{2-}$), trace elements (As, and $Fe^{2+}$), and physicochemical parameters (pH, Temperature, electrical conductivity (EC), total dissolved solids (TDS), and Turbidity) 4) to comprehend the water quality index and hydrogeochemistry in groundwater samples from the research region in the Thal Desert in order to implement the appropriate rehabilitative steps. This research not only unveils the challenges associated with groundwater contamination but also emphasizes the need for proactive measures. The intricate balance between economic growth and environmental preservation requires a paradigm shift towards sustainable groundwater management. By spotlighting a specific case study, this paper serves as a catalyst for heightened global awareness, research endeavors, and pragmatic interventions to safeguard this invaluable resource, ensuring its sustainable use for the prosperity of economies and the well-being of communities worldwide.

## Study area

The Thal desert is located in Pakistan's Punjab province between 30˚ and 32˚30' North and between about 71˚ and 72˚ East. The region is a sub-tropical sandy desert that extends 190 miles, with a maximum width of 70 miles. The Thal Desert experiences extremely hot summers, with typical temperatures averaging in June-July ~35˚C and falling in December-January to 10˚C. The north and west experience lower annual average temperatures, which range from 24˚C to 28˚C. Less than 350 mm of rainfall is recorded annually in the majority of the area. The average annual rain fall in southern margin was reported to 150 mm [33] while in the northern margins it has slight higher values up to 618 mm [3]. The area is bordered to the North by the Salt Range Piedmont, to the West by the flood plains of the Indus River, and to the East by the flood plains of the Chenab and Jhelum Rivers. This area is subdivided into the districts of Muzaffargarh, Bhakkar, Layyah, Jhang, Khushab, and Mianwali. The study area comprises of Muzaffargarh city (Fig 1a). The Indus River supplies all of the wind-reworked sand for the Thal Desert dunes, which are upstream of the orogenic front and delivers us a concise compositional trace of detritus mostly produced by the fast degradation of the Himalayan syntaxis in the Quaternary [2, 34]. The Quaternary eolian and fluvial deposits are >350 m thick in the southern parts, but considerably thicker in the central and adjacent portion of the desert, where rolling sand plains or low sand dunes alternate with narrow lowlands of cultivated areas [22] (Fig 1b). Mesolithic artifacts were recently found at the crest of the dunes, proving that eolian sand deposits were before the Holocene [2, 35]. The subsurface alluvial deposits are mainly comprised of lateral layers of fine to coarse sand, with a small amount of gravel and a few scattered mud lenses. The distribution of grain sizes is uneven throughout, suggesting initial deposition by the continuously evolving paleo-Indus and/or nearby

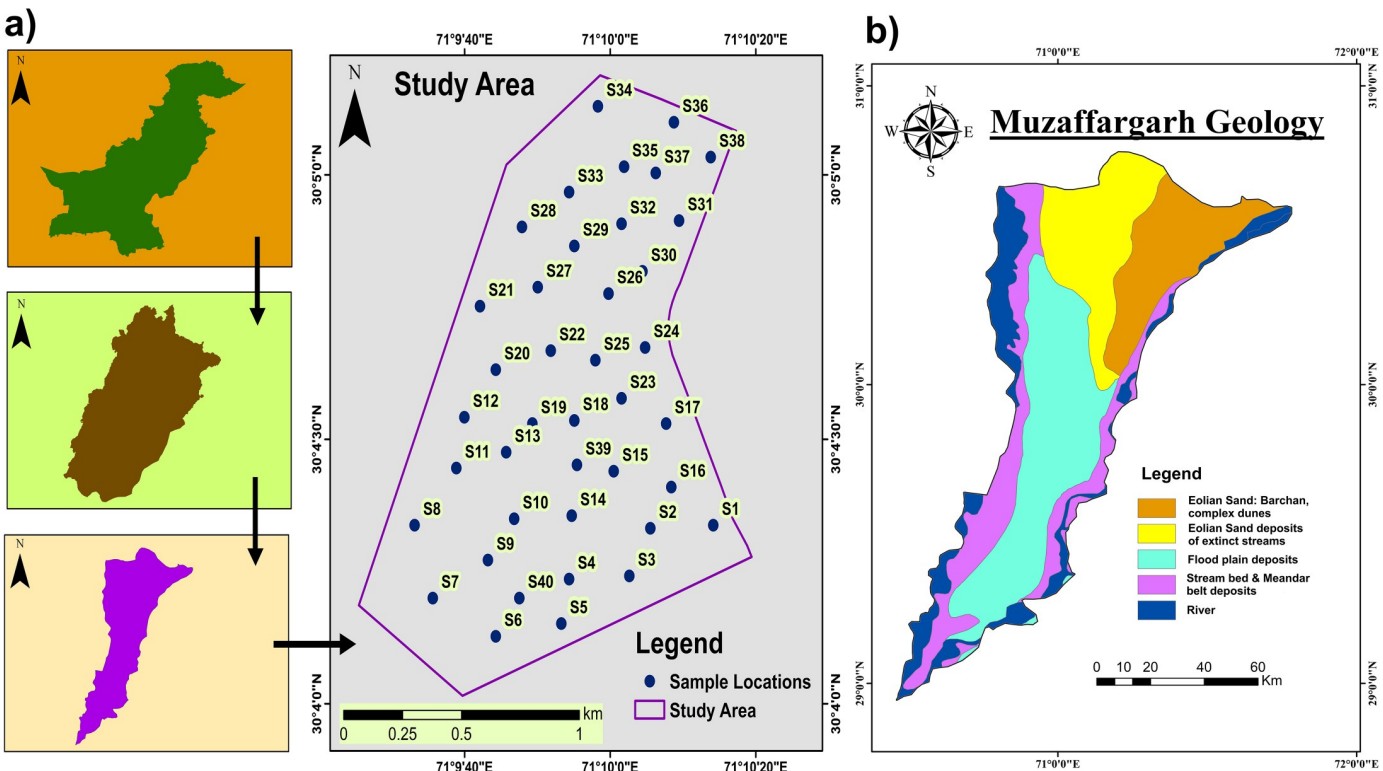

**Fig 1.** a) The study area map providing an overview of the locations where the samples were collected. b) A blueprint of the study area's geology.

tributaries, with coarser deposits occurring in the north closer to the Salt Range [36]. The southernmost portion of the currently active Indus River floodplain is more than 20 kilometers wide [33, 37]. River water and rainfall quickly replenish the Thal Doab aquifer, which is made up of Quaternary eolian and alluvial sediments with local mud lenses [32, 38].

The hydrogeological characteristics significantly influence the supply and replenishment of the aquifer. Reworked loess deposits are considered to be the source of finer deposits, such as clayey and silty contents found below the surface [39–41]. The coarser sediments, like sand and gravel, have a higher permeability than the finer sediments, even though both types of sediments are porous. As a result, the finer sediments serve as an aquitard, hindering water movement and increasing the salinity of groundwater naturally [42, 43]. The intermixing of such fine sediments causes hindrances to groundwater flow in various parts of Thal Doab. The higher elevation and coarse grained sediments of the Indus River play an important role for regulating the drainage and hydrology of Thal Doab, which subsequently stimulates aquifer replenishment. The drainage system fed by numerous off-takes of the Indus River, traverses Thal Doab which is a recharging source of alluvium aquifer through a natural gradient. Deterioration of groundwater quality can also be observed; however, its quality tends to improve in closer proximity to the riverbeds [44].

## Methodology

### Field data collection

The field site access for this study was approved by the Institute of Geology, University of the Punjab. Groundwater samples were collected from 40 different places in order to perform

quantitative, qualitative, and semiqualitative microbial analysis. On the topographic survey sheet, the wells locations were accurately noted. Latitudes and longitudes were also recorded during the survey. A portable global positioning system was used to determine the exact position of the groundwater samples. All specimens for physicochemical and hydrochemical assessment were randomly taken from hand pumps at depths ranging from 11–28 m in 1-liter polyethylene bottles. Prior to sampling, all wells were pumped for a period of 5–10 minutes at flow rates ranging from 3–200 $m^3h^{-1}$ to evacuate any remaining stagnant water from the casing and storage system. One portion of the samples was acidified and 2 to 10 ml of 35% $HNO^3$ was added for significant cations in order to preserve them. Another portion of the specimens was secured for analysis of significant anions in the aquifer. Using a combination of pH, EC meters, and thermometers, field measurements of physical attributes including temperature, pH, TDS, and EC were taken. Through a sensory evaluation approach, the color, taste, and odor of water samples were assessed. Similarly, for the assessment of semiqualitative bacteriological contamination, groundwater specimens were taken discretely in sterilized and clean testing bottles. The sampling process was carefully monitored to avoid any unintentional contamination.

## Physio-chemical analysis of groundwater samples

The physical, chemical, and microbiological characteristics of the water specimens were analyzed using the laboratory standard procedures. Temperature, pH, color, odor, taste, TDS, EC, turbidity, arsenic, hardness, sodium, potassium, iron, magnesium, calcium, fluoride, chloride, bicarbonate, nitrate, sulfate, and microbial were all evaluated in each water sample. TDS was determined experimentally by taking 50 ml of groundwater specimens in a pre-weighed beaker kept in a water bath. Prior to starting measurements for EC, samples were properly agitated and allowed to stabilize until the air bubbles were eliminated. EC meter, HACH-44600, USA, was used to measure EC. 0.01 M of standard potassium chloride solution, at 25°C of constant temperature was used to standardize the EC meter. Formazin polymer was utilized as the baseline turbidity standard suspension and the nephelometric technique was employed to determine the turbidity of the samples (Lamotte, model 2008 USA). The instrument scale provided a clear indication of the turbidity. Merck Test Kit, Germany (0.01–0.5 mg/1) is employed to evaluate the amount of arsenic present in groundwater samples. Arsenic-Ill and V concentrations are assessed semi-quantitatively by comparing the reaction zone of the analytical test strip to the color scale fields. Atomic Absorption Spectrophotometer (Perkin-Elmer A Analyst 600 Graphite Furnace) is utilized to verify the results of the field testing on the levels of arsenic in the groundwater. The Photometric Phenanthroline Technique was implemented to analyze the iron. On a HITACHI Spectrophotometer, Model L -1100, the concentration of iron was quantified at 510 nanometers. Titration was used to measure quality factors including alkalinity, chlorides, bicarbonates, and calcium. If proper inhibitors are used in the hardness titration and intervening metals are present in the calcium titration at non-interfering ratios, magnesium ($Mg^{2+}$) can be calculated by subtracting calcium ($CaCO_3$) from hardness. Ion-selective electrodes (ISE) were used to measure the ions of fluoride and nitrate (Ion Meter/Cyber Scan pH). Fluoride was also analyzed using Spectrophotometer technique 8029, SPADNS (Hach). The suitability of potassium ($K^+$) and sodium ($Na^+$) in aquifer specimens is assessed using a flame photometer (PFP7, UK). A UV-vis spectrophotometer operating at a wavelength of 420 nm was used to measure the presence of sulfate.

## Microbial analysis

Additionally, a semi-qualitative test for the microbial analysis of groundwater samples is performed using a Microbiological Testing Kit. By using the Multiple Tube Fermentation

Method, microbiological testing for Total Coliforms, Fecal Coliforms, and E. Coli was performed. Total Coliforms serve as general indicators of bacterial contamination in water, with their presence suggesting potential fecal contamination. Fecal Coliforms, a subset of Total Coliforms, specifically indicate fecal contamination and are commonly associated with the intestinal tracts of warm-blooded animals. E. Coli, a specific type of Fecal Coliform, is a key indicator of fecal pollution and is often used as a marker for the potential presence of harmful pathogens in water. After 24 to 48 hours of incubation at 350°C, each inoculated tube was checked for the generation of gas and acid. The tubes containing gas and acid are designated as presumed "positive". A test is considered negative if, after 48 hours of incubation, no gas or acid has been generated.

## Interpolation

In order to describe geographic events spatially precisely, geo-statistical interpolation algorithms are helpful. In order to determine the susceptible areas for groundwater resource management and sustainability, the spatial distribution of drinking groundwater quality values was plotted using ArcGIS software employing the inverse distance weighted (IDW) interpolation approach [45–48]. The spatial variation of the eighteen water quality attributes was mapped in this study using the IDW geostatistical interpolation approach. When evaluating values in uncharted territories, this approach was adopted because of its capability to represent events by taking into consideration both the spacing and the amount of variance between known data values [49]. Additionally, IDW makes the assumption that the spacing or orientation between sampling locations reveals a spatial relationship that may be employed to account for surface variation. The output value for each site is calculated by fitting a mathematical function to a particular number of points within a defined radius using the IDW tool, which assumes an unknown constant mean [28, 47].

## Graphical visualization to analyze the chemistry of groundwater

The utilization of a diverse array of graphical representations is a well-established methodology aimed at providing a comprehensive understanding of hydrochemical data in a specific study area. These visual tools are expertly crafted to depict the intricate relationships between distinct chemical parameters and the overarching quality of water resources. By overlaying hydrochemical data onto these purpose-built diagrams, researchers are granted invaluable insights into the origins, mechanisms, and potential implications of diverse chemical constituents on water quality within the geographic scope of the study. This knowledge is pivotal in steering effective water management strategies, shaping policy decisions, and identifying critical areas warranting further investigation to unlock a deeper comprehension of the nuanced interactions between chemical elements and the natural environment. Noteworthy among these graphical techniques are the Hill-Piper Trilinear diagram, Durov diagram, Gibbs diagram, Stiff plot, scatter plot matrices, pie charts, and the Wilcox diagram, each assuming a pivotal role in unraveling the complexities of groundwater hydrochemistry and gauging its suitability for a wide range of applications.

In the realm of hydrogeochemical investigations, a variety of graphical methodologies is harnessed to scrutinize the composition and quality of groundwater resources. The Hill-Piper Trilinear diagram stands as a prime example, categorizing groundwater into six distinct compositional types predicated on ion concentrations. This diagram, composed of an apex diamond and two subordinate triangles, elegantly represents the anions and cations governing the elemental composition of groundwater [23, 49]. A complementary graphical tool, the

Durov diagram, artfully portrays the hydrogeochemical attributes of diverse samples by highlighting the proportion of major anions and cations present in each.

Typically crafted using software like Grapher, the Piper and Durov diagrams offer profound insights into the intricate nuances of groundwater hydrochemistry. Exploring the interplay between lithology and hydrochemistry within aquifers often involves the application of the Gibbs diagram. By relating the geological formations to the chemical composition of water, this graphical method serves as an instrumental guide. Another pivotal technique, the Stiff plot, visually showcases the chemical makeup of water samples. The size and shape of the polygon in the Stiff plot offer key insights into the unique chemical characteristics of each sample.

A statistical technique for determining the strength and direction of a linear relationship between two continuous parameters is the Pearson correlation coefficient. Using the Past4.13 tool, the Pearson correlation coefficient is calculated. Further augmenting the investigative arsenal are scatter plot matrices generated via Geographic Information Systems (GIS), which prove invaluable in identifying spatial patterns and relationships in datasets. In parallel, circular pie charts derived from GIS data provide a coherent and intuitive representation of the collected information. Finally, the Wilcox diagram emerges as an indispensable tool for evaluating the viability of groundwater for irrigation. Plotting the sodium adsorption ratio (SAR) against the electrical conductivity (EC) of water samples, this diagram delineates distinct regions representing varying degrees of suitability for both drinking and irrigation purposes. Samples positioned in the lower-left quadrant of the diagram (characterized by low SAR and low EC) are generally deemed suitable for irrigation, while those in the upper-right quadrant (reflecting high SAR and high EC) are typically not suitable.

## Water quality index (WQI)

The assessment of water quality was conducted by computing the WQI using the recommended drinking water quality standard of the World Health Organization [50]. WQI was determined using the weighted arithmetic method, which was first introduced by [51] and subsequently refined by [52]. The formula for calculating the WQI using the weighted arithmetic method is provided as follows:

$$\text{WQI} = \frac{\sum_{i=1}^{n} W_i Q_i}{\sum_{i=1}^{n} W_i} \tag{1}$$

Where $n$ = number of variables or parameters, $Wi$ = unit weight for the ith parameter, $Qi$ = quality rating (sub-index) of the ith water quality parameter.

For the purpose of our study, we have utilized the WQI classification system developed by [52], as well as those developed by [49, 53, 54]. This classification system has been used to determine the water quality and its suitability for various uses. The results of this classification are presented in Table 3 and serve as a reference for our study.

## Results

### Geospatial analysis of water quality parameters

Assessing the quality of the groundwater is crucial as it is the primary determinant of its appropriateness for drinking, domestic, agricultural, and industrial uses [55]. Water samples were taken from 40 distinct locations at varying depths from 11m to 28m, with an average depth of 16m, as illustrated in Fig 2a. The outcomes of the groundwater sample physiochemical analysis are interpolated in Fig 2a–2v. The outcomes were evaluated in comparison to the WHO's water quality guidelines [50, 56, 57]. Many of the 40 samples of groundwater taken from

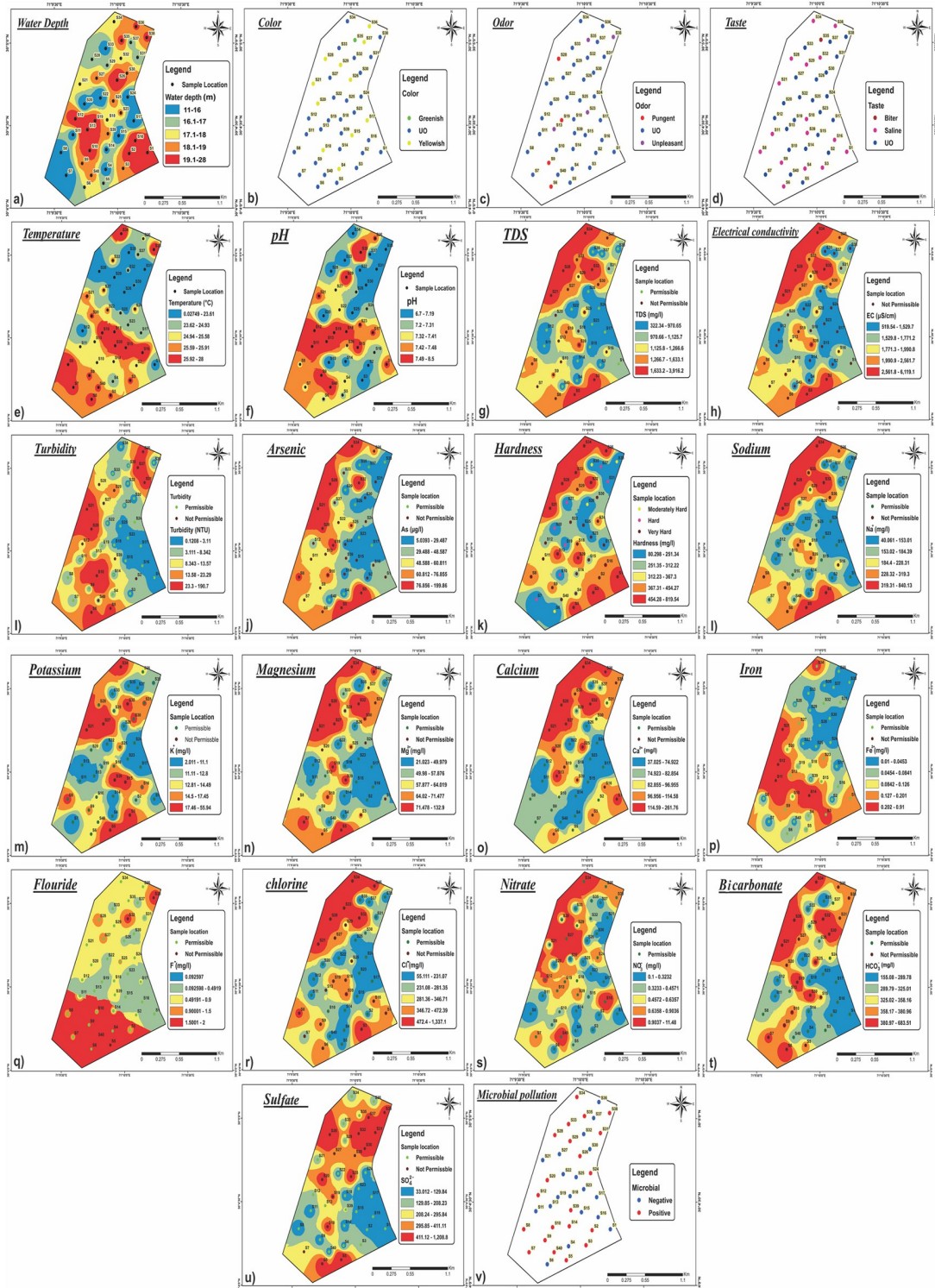

**Fig 2.** IDW maps of the research region illustrating the geographical distribution of physicochemical parameters a) Water depth b) Color c) Odor d) Taste e) Temperature f) pH g) TDS h) EC i) Turbidity j) Arsenic k) Hardness l) Sodium m) Potassium n) Magnesium o) Calcium p) Iron q) Fluoride r) Chlorine s) Nitrate t) Bicarbonate u) Sulfate v) Microbial pollution.

**Table 1. Comparative analysis of groundwater quality and statistical characterization of physicochemical parameters in groundwater samples from Thal desert (n = 40) [50, 56, 57].**

| Sr. no. | Parameters | Units | Min | Max | Mean | WHO | NSBL | NSBL % |
|---------|-----------|-------|-----|-----|------|-----|------|--------|
| 1 | pH | - - | 6.7 | 8.5 | 7.34 | 8.5 | 0 | 0 |
| 2 | TDS | mg/L | 322 | 3920 | 1327.8 | 1000 | 24 | 60 |
| 3 | EC | µS/cm | 519 | 6125 | 2082.8 | 400 | 40 | 100 |
| 4 | Turbidity | NTU | 0.1 | 190.7 | 15.6 | 5 | 15 | 37.5 |
| 5 | As | µg/L | 5 | 200 | 56.2 | 10 | 31 | 77.5 |
| 6 | $Fe^{2+}$ | mg/L | 0.01 | 0.91 | 0.14 | 0.3 | 4 | 10 |
| 7 | $Na^+$ | mg/L | 40 | 841 | 240.9 | 200 | 16 | 40 |
| 8 | $K^+$ | mg/L | 2 | 56 | 14.4 | 20 | 8 | 20 |
| 9 | $Mg^{2+}$ | mg/L | 21 | 133 | 61.8 | 50 | 23 | 57.5 |
| 10 | $Ca^{2+}$ | mg/L | 37 | 262 | 96.1 | 100 | 17 | 42.5 |
| 11 | Hardness | mg/L | 80 | 820 | 352.3 | - - | - - | - - |
| 12 | $F^-$ | mg/L | 0.09 | 102 | 3.1815 | 1.5 | 3 | 7.5 |
| 13 | $Cl^-$ | mg/L | 55 | 1338 | 362.95 | 250 | 20 | 50 |
| 14 | $NO_3^-$ | mg/L | 0.1 | 11.5 | 0.791125 | 50 | 0 | 0 |
| 15 | $HCO_3^-$ | mg/L | 155 | 684 | 342.05 | 350 | 19 | 47.5 |
| 16 | $SO_4^{2-}$ | mg/L | 33 | 1210 | 286.75 | 250 | 15 | 37.5 |

Note: NSBL is number of samples below the limit.

different places in the research region at shallow depths (11–28 m) were reported to be of an undesirable yellow or greenish color (Fig 2b). Only eight shallow handpump wells (11–28 m) were reported to have an unpleasant or pungent odor (Fig 2c). Additionally, the research region's groundwater specimens from 17 of the 40 collected samples tasted salty or bitter (Fig 2d). The groundwater temperature remains constant across the research region, ranging from 0 to 28 ˚C, with an average temperature of 24.52 ˚C (Fig 2e). The pH of groundwater samples of the study area fluctuates between 6.7 and 8.5 (average 7.34) (Fig 2f). Because of the $CO_2$ loss and mineral salt accumulation and penetration, groundwater is considered to be moderately alkaline in all climates. With an average of 1327.875 mg/L, TDS ranges from 322 to 3920 mg/L (Table 1, Fig 2g). The maximum acceptable value of 1000 mg/L is exceeded at around 60% of sampling locations, making them not suitable for drinking. The research area's EC has an average value of 2082.8 µS/cm and a variation of 519 to 6125 µS/cm, which indicates high mineralization of weathered material from sub-Himalaya and the influence of human activity (Fig 2h). Turbidity in water is caused by suspended particles, including fine organic and inorganic particles, clay, silt, soluble colored organic compounds, plankton, and other microorganisms [58]. The range of turbidity values is 0.1–190.7 NTU, with an average of 15.66 NTU (Fig 2i; Table 1). The WHO-permitted standard of 5 NTU is exceeded by almost 37.5% of the total 40 water samples. 62.5% of the remaining groundwater results are under WHO guidelines (Fig 2i). The World Health Organization (WHO) recommends a 10 mg/L arsenic content in drinking water. In Muzaffargarh (City), Pakistan, the average concentration of arsenic in the groundwater is 56.25 µg/L (Table 1), which is obviously above permissible limits. Only seven samples (S-4, S-17, S-23, S-26, S-29, S-35, S-37, S-38, and S-40) out of 40 were identified with arsenic concentrations that were considerably below the permissible limit (Fig 2j). The average value of hardness is 352.4 mg/L with the lowest and highest quantities of 80 to 820 mg/L (Fig 2k).

Between 40 and 841 mg/L of sodium are present in the study region. According to (Fig 2l) 16 (40%) out of the total specimens collected exhibited contamination levels above the WHO standards. The degradation and percolation of clay minerals and sodium feldspars from the study region contributed to the elevated $Na^+$ in the groundwater there. $Na^+$ content rises in the research region (Fig 2l) from NW to S in relation to groundwater flow and longer residence times because of minimal inclination. The reason for the groundwater sample's highest $Na^+$ content may be associated with residential pollutants and soil leaching. Potassium ($K^+$) is always present in lower concentrations than other cations, and it is often less than 10 mg/L in natural waters. 2 mg/L to 56 mg/L of $K^+$ is present in groundwater (Fig 2m). Out of 40 groundwater samples, 8 samples exceeded the WHO limit of 20 mg/L. Increased fertilizer usage and subsurface leaching might be to cause of the highest recorded amount of $K^+$ at S-27 (56 mg/L). The range of magnesium ($Mg^{2+}$) concentration in groundwater is 21 to 113 mg/L (Fig 2n). 57.5% of the groundwater samples exceeded the WHO guidelines. Carbonates, gypsum, and ferromagnesian minerals are sources of magnesium, which increases the amount of magnesium in groundwater. Fig 2o depicts a spatial fluctuation map of calcium ($Ca^{2+}$). The average $Ca^{2+}$ concentration in the study region is 96.125 mg/L, with a range of 37 to 262 mg/L (Fig 2o). The high concentration of calcium in the basin's groundwater indicates feldspar in the region's rocks as the primary source. Groundwater has an average iron content of 0.1385 mg/L, with a range of 0.01 to 0.91 mg/L. In terms of water quality identified with $Fe^{2+}$, 4 of the 40 sites exceeded WHO guidelines, as shown in Fig 2p.

The average groundwater fluoride content is 3.1815 mg/L, but the range is 0.09 to 102 mg/L. As seen in Fig 2q the 3 sites surpassed WHO guidelines for water quality as indicated by the symbol $F^-$. Fluoride concentrations exceeding WHO guidelines were present in 7.5% of water samples. 55 to 1338 mg/L of chloride ($Cl^-$) are the minimum and maximum quantities with an average value of 362.95 mg/L. About 52.5% of the samples exceeded the WHO guidelines (250 mg/L) as shown in Fig 2r. Untreated sewage, domestic waste, and fertilizers, which are examples of human and natural sources, may be responsible for the elevated $Cl^-$ levels detected in the samples. The concentration of nitrate ($NO_3^-$) ranges from 0.1 to 11.5 mg/L with an average value of 0.7911 mg/L (Fig 2s). Groundwater always has a natural content of $NO_3^-$ less than 10 mg/L. Zones of high proportions were recorded, reflecting the usage of fertilizers high in nitrogen as well as human waste. In the Muzaffargarh Thal desert, bicarbonate ($HCO_3^-$) concentrations range from 155 mg/L, which is the minimum, to 684 mg/L, which is the maximum (Fig 2t). The abundant supply of $CO_2$ from rainwater recharge via the shallow soil zone into aquifers and organic matter decomposition are the causes of the considerable concentrations of $HCO_3^-$ found in groundwater samples. The concentration of sulfate ($SO_4^{2-}$) in groundwater varies from 33 to 1210 mg/L. The maximum permissible sulfate concentration for drinking water is 250 mg/L, and 15 water samples with excessive sulfate concentration exceeded the WHO guidelines (Fig 2u). Human activities and fertilizers contribute to the basin's high sulfate content.

## Microbial analysis

The microbial contamination analysis of these samples reveals that 21 (52.5%) of these wells are polluted with coliforms and fecal coliforms (Fig 2v). Fecal organics identified in the groundwater indicate contaminated sources of water, which may also cause the mobilization of arsenic by causing local redox zonation in the groundwater. The contamination of shallow aquifers in the region is attributed to inadequate sanitation facilities and the absence of lined sanitation systems. Factors contributing to microbial contamination include unlined sanitation practices, the discharge of organic waste, anthropogenic activities, and the utilization of

open ponds for the disposal of human and animal waste. This may also contribute to the release of arsenic through iron reduction. The correlation between oral ingestion of water contaminated with fecal matter and waterborne diseases is well-established. The prevalence of microbial contamination in most wells has led to the widespread occurrence of diseases such as gastroenteritis, typhoid fever, cholera, dysentery, infectious hepatitis, bacillary diarrhea, and other bacterial infections. Fecal organics in the water supply have particularly resulted in prevalent issues like diarrhea, gastroenteritis, and cramps among the local population.

Municipal water supplied to the public is frequently tainted with infectious microorganisms and hazardous chemicals. During the monsoon season, when surface water contamination is heightened, there is a surge in epidemic-like outbreaks of gastroenteritis and diarrhea due to the inadequate sanitation system. This poses a significant risk to the lives of numerous individuals, especially children, the elderly, and women. The absence of an effective municipal water supply system in these areas exacerbates the contamination issue. Groundwater is compromised due to a deficient sewage line system, leading to waterborne diseases in the locality. Insufficient sanitation facilities result in the drainage of sewage water into stagnant ponds, where it gradually mixes with subsurface groundwater, causing microbiological contamination. Efforts are needed to address these sanitation challenges and protect the local population from the adverse health impacts associated with contaminated water sources.

## Hydrogeochemical evaluation of groundwater through graphical projections

The Piper diagram serves as an effective means of categorizing groundwater into six distinct types, including $Ca^{2+}$-$HCO_3^-$, $Na^+$-$Cl^-$, mixed $Ca^{2+}$-$Mg^{2+}$-$Cl^-$, $Ca^{2+}$-$Na^+$-$HCO_3^-$, $Na^+$-$HCO_3^-$, and $Ca^{2+}$-$Cl^-$, based on its chemical composition. After analyzing the diagram, it becomes evident that a significant proportion of the samples falls within the mixed $Ca^{2+}$-$Na^+$-$HCO_3^-$ and $Ca^{2+}$-$HCO_3^-$ categories, which indicates that there is a high concentration of calcium and sodium-bearing minerals and salts. Conversely, the mixed $Ca^{2+}$-$Mg^{2+}$-$Cl^-$ and $Na^+$-$Cl^-$ categories contain fewer samples as depicted in Fig 3a. The hydrochemical analysis of the samples reveals that the groundwater contains cations in the order of $Na^+$> $Mg^{2+}$> $Ca^{2+}$> $K^+$,

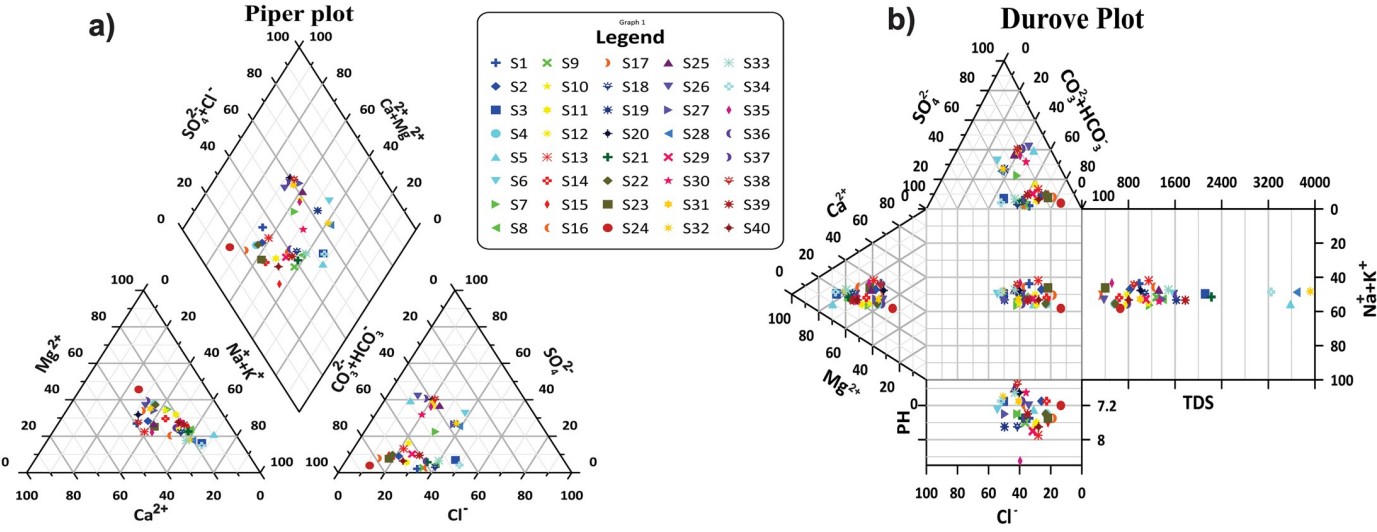

**Fig 3.** a) Exploring water quality trends in study area with Piper Diagram analysis b) Exploring water quality trends in study area with Durov Diagram analysis.

and anions in the order of $HCO_3^- > SO_4^{2-} > Cl^-$. This finding indicates that bicarbonate, sodium, and chloride ions are the dominant ionic species in the groundwater due to weathering processes and ion exchange.

The Durov diagram is a useful tool for hydrogeochemical data analysis, revealing that the mixed type was the most common groundwater type (Fig 3b). In the categorization of samples, a limited number fell within the C field, indicating the presence of $Mg^{2+}$. Conversely, there was notable diversity observed in the anion composition of groundwater samples, with the majority classified as belonging to the bicarbonate E-type, as depicted in Fig 3b. These samples were mainly composed of freshwater with TDS levels below 1000 mg/L and located in the aquifer recharge region. The remaining samples were classified as either brackish water with sulfate or mixed water types, influenced by factors like evaporation, water-rock interactions, and anthropogenic activities.

The Gibbs diagram is a useful tool for studying factors that influence groundwater chemistry, including precipitation, rock weathering, and evaporation. The majority of groundwater samples studied fell into the rock dominance and evaporation regions, indicating the critical roles of rock weathering and evaporation in shaping groundwater chemistry (Fig 4). Mineral groundwater enrichment was primarily attributed to rock weathering, which was facilitated by an extended period of contact between the rock and water. Anthropogenic activities like agricultural fertilizers and irrigation return flows can also affect groundwater chemistry by increasing the concentrations of $Na^+$ and $Cl^-$ and causing an increase in total dissolved solids. Considering these factors helps to achieve a more comprehensive understanding of the underlying mechanisms driving variations in groundwater chemistry.

By analyzing the stiff plots, we can identify the dominant geochemical processes affecting the groundwater quality in the region. We can also use the information to design appropriate water treatment strategies to address any issues related to water quality. The resulting plots show different diagonal lines, each representing a different geochemical process. A diagonal line of $Ca^{2+}$ and $HCO_3^-$ ions indicates the dissolution of calcite, in the aquifer system.

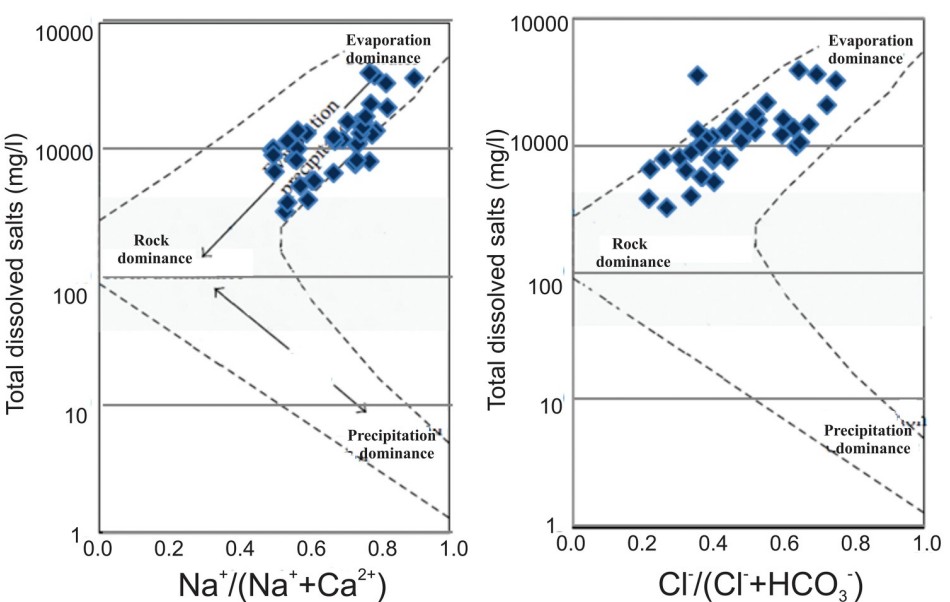

**Fig 4. Groundwater quality assessed through Gibbs Diagrams: (a) Na/Na$^+$ + Ca$^{2+}$ mg/L versus Log TDS mg/L, and b) Cl$^-$/Cl$^-$ + HCO3- mg/L versus Log TDS for the study area.**

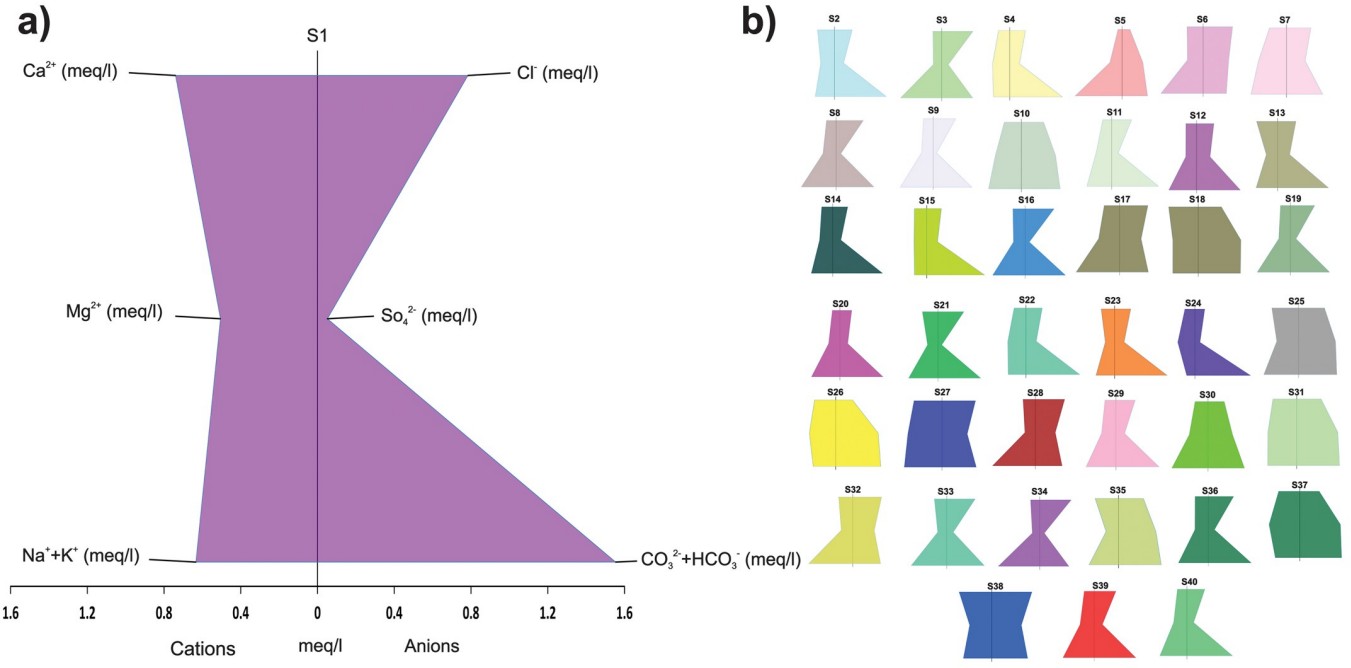

**Fig 5.** a) Comparison of different parameters using the Stiff Plot for enhanced visualization and analysis of S1. b) Stiff Plots of 39 samples showcasing the distribution of data across multiple parameters.

Similarly, a diagonal line of $Na^+$ and $Cl^-$ ions suggests the influence of evaporation as shown in Fig 5a and 5b.

A straightforward method to express the linear relationship between two variables is through the Pearson correlation coefficient. It spans from -1 to +1 and comprises positive, no, and negative correlations, enabling researchers to understand the degree and direction of links within their data. The findings indicate Pearson correlation coefficients between several water quality parameters including temperature, pH, TDS, EC, turbidity, and concentrations of various ions and compounds like As, $Fe^{2+}$, $Na^+$, $K^+$, $Mg^{2+}$, $Ca^{2+}$, $Cl^-$, $F^-$, $NO_3^-$, $HCO_3^-$, $CO_3^{2-}$, and $SO_4^{2-}$ in the water (Fig 6). These correlation coefficients shed light on potential connections between various parameters of water quality. Evaluating these relationships can help find potential sources of contamination, natural alterations, and underlying processes that regulate the behavior of these variables in aquatic systems. When analyzing these relationships, it is essential to examine both natural and human-induced factors, as well as carry out more research, including field investigations and laboratory tests, to confirm and expand on these findings. The strong correlations between various parameters are attributed to common sources, hydrogeological and geomorphic conditions, anthropogenic influences, and chemical interactions. Natural geomorphic conditions, mineral dissolution, and weathering, release ions into water bodies, affecting their concentrations. Anthropogenic activities like domestic effluents, and industrial processes introduce pollutants into water sources, influencing the observed correlations. Moreover, chemical interactions between ions lead to complex equilibria and shifts in concentrations, further impacting correlation patterns.

A scatter plot matrix visualizes the correlations between numerous physicochemical parameters by showing one parameter against all others in a matrix manner (Fig 7). The strength and

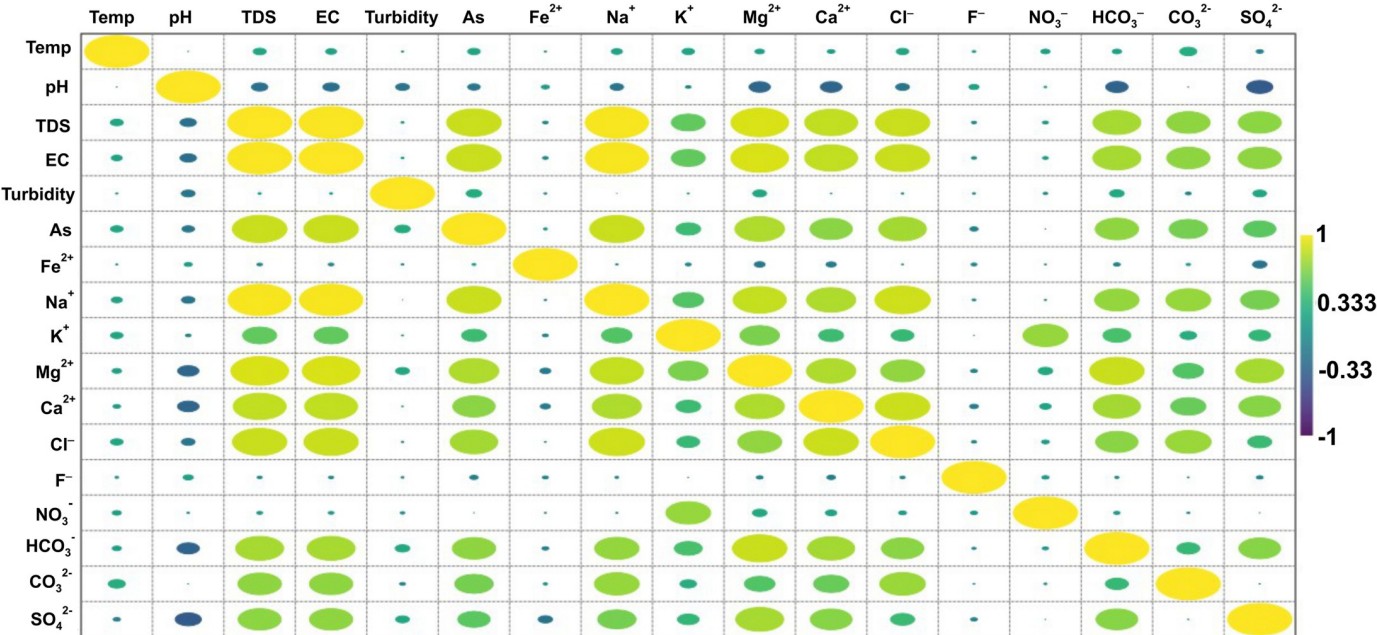

**Fig 6. Visualization of relationships with different parameters using the Pearson correlation matrix.**

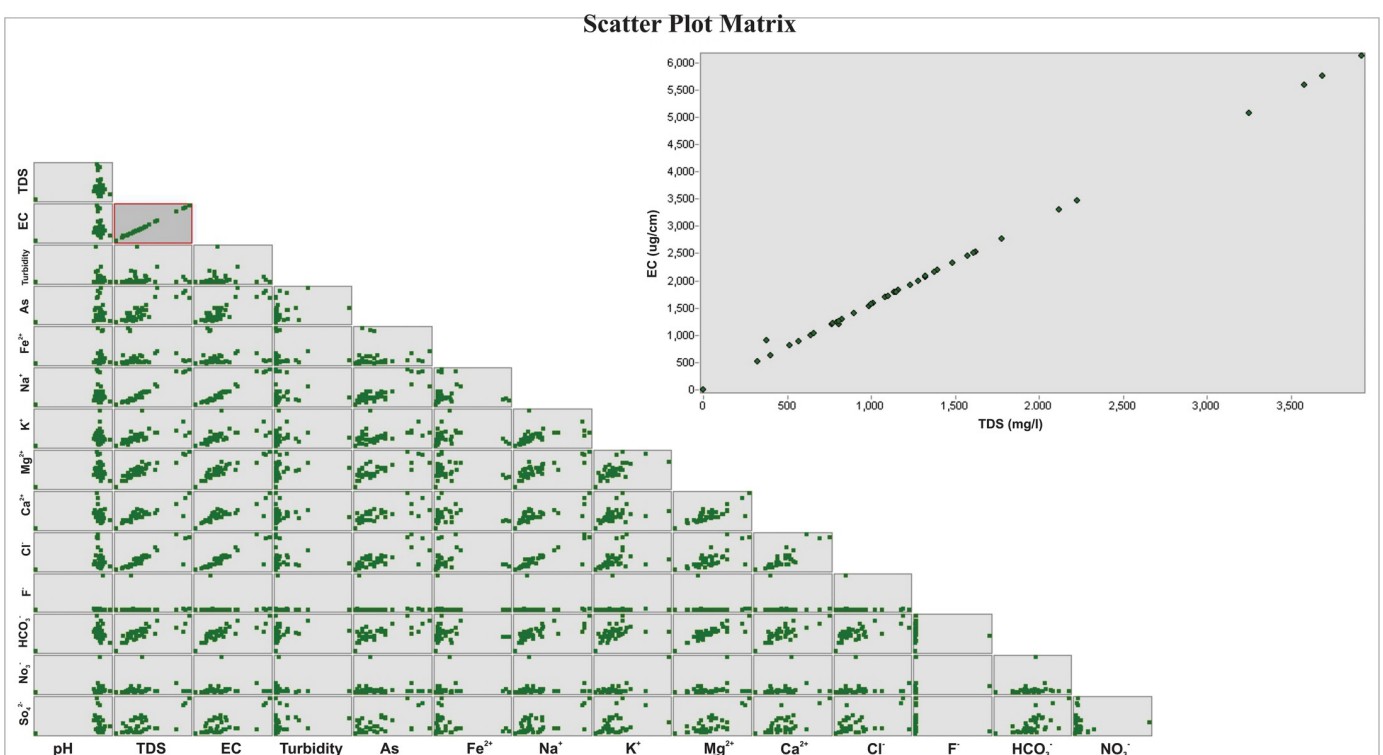

**Fig 7. Scatter plot matrix showing a graphical depiction of the correlation between all parameters.**

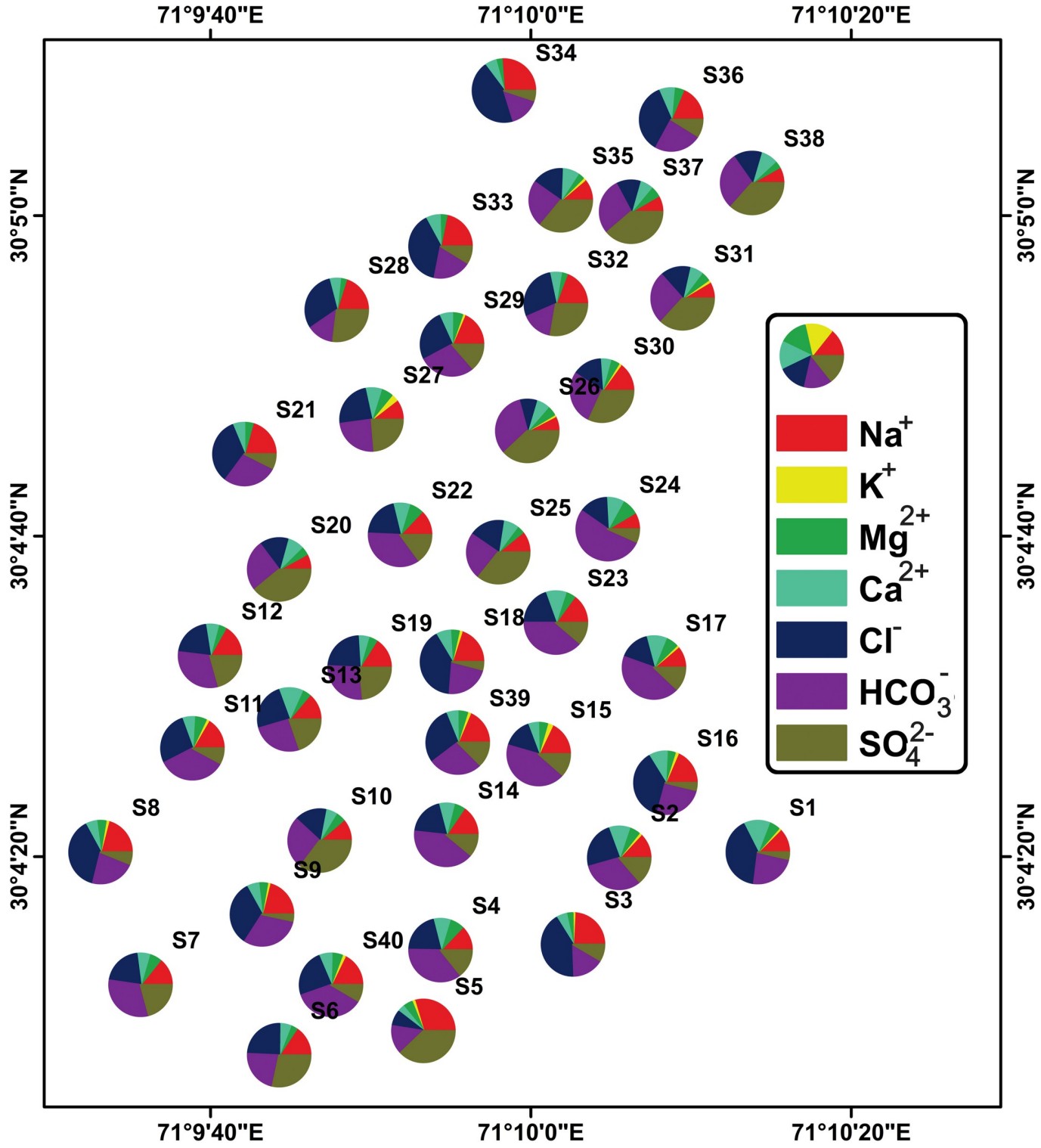

**Fig 8. Graphical depiction of the percentage of each category in the dataset using a pie chart.**

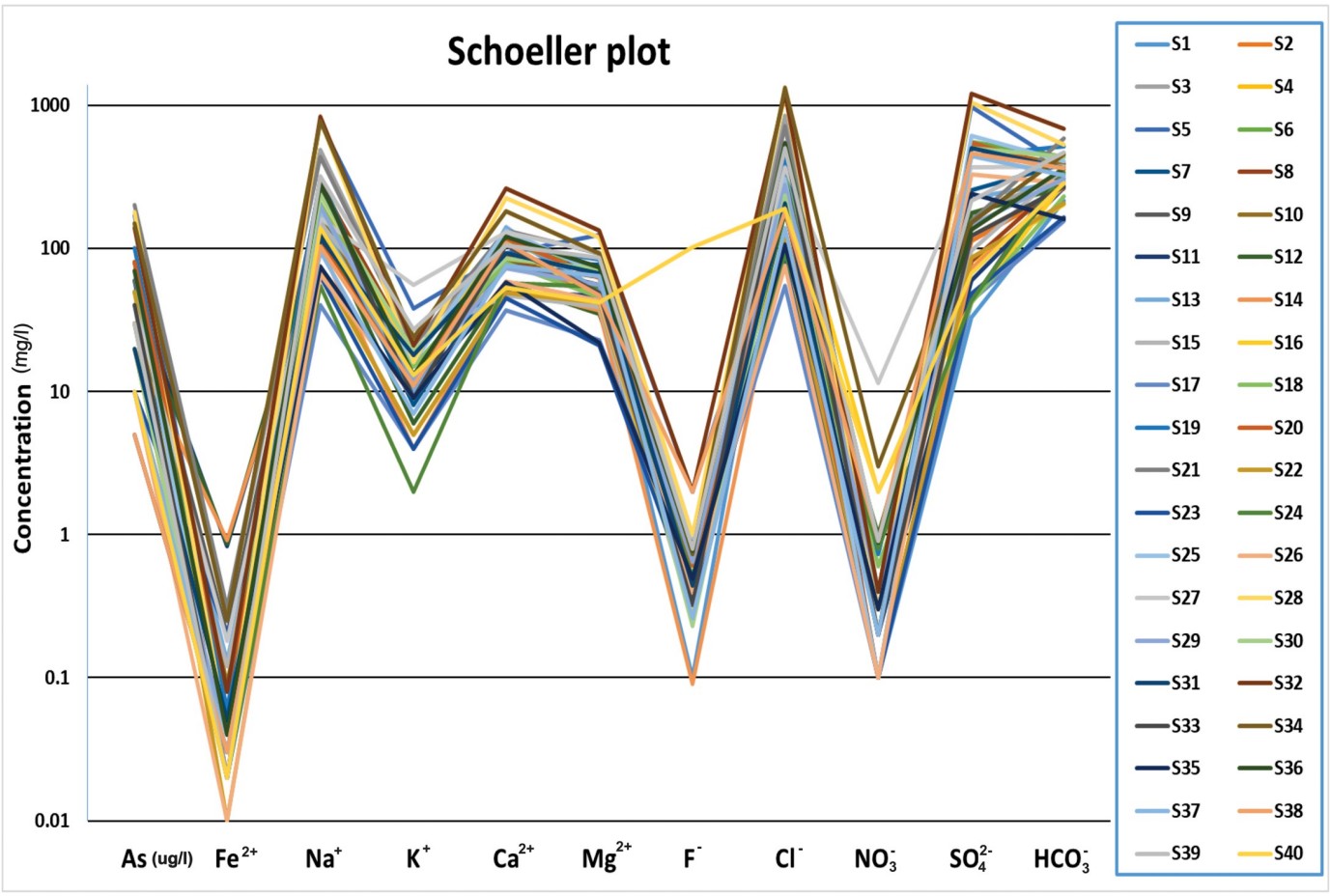

**Fig 9. Scholler plot depicting the fluctuations of variables in groundwater samples based on the concentration.**

direction of the relationship between various variables are shown by patterns and trends that can be identified by analyzing the scatter plot matrix.

In hydrogeology, pi charts are a valuable tool that may be used to determine the primary characteristics impacting water quality in a specific sample and to compare the composition of other samples. The pi chart of S1, for example, shows that the concentration of groundwater samples follows the order $Cl^- > HCO_3^- > Ca^{2+} > Na^+ > Mg^{2+} > SO_4^{2-} > K^+$ as shown in Fig 8. These charts may be used for finding spatial patterns in all samples. These charts help hydrogeologists discover trends and patterns in water quality data, as well as acquire insights into the subsurface hydrogeological processes that influence the observed patterns.

The results obtained from the Schoeller diagram indicated a consistent trend in the curve of water samples in the study area, which strongly suggests that the groundwater sources in the region are relatively homogeneous (Fig 9). This further implies that the chemical characteristics of the water samples are similar, and they likely originate from the same hydrogeological system. These findings have practical implications in identifying potential sources of contamination, developing effective water management strategies, and ensuring the sustainable use of groundwater resources in the region.

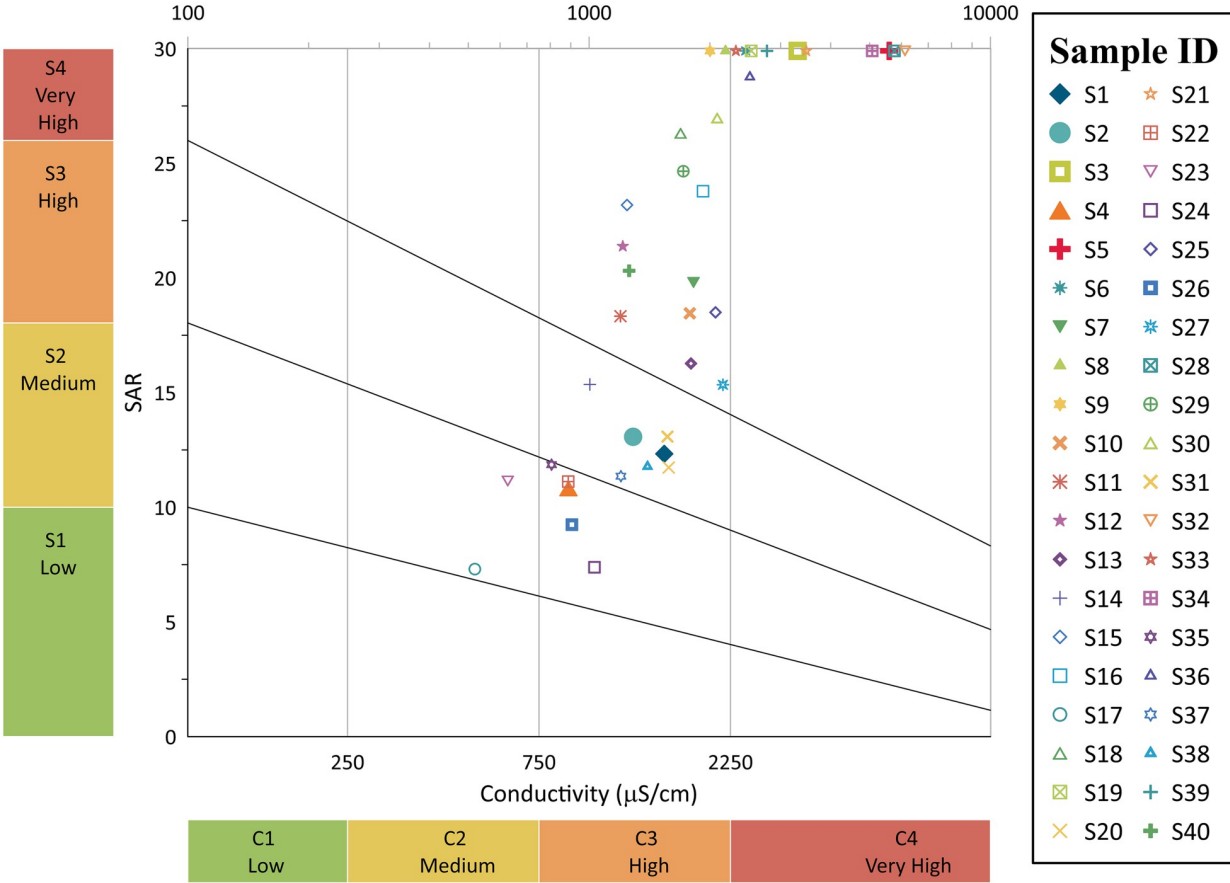

**Fig 10. Visual representation of the chemical quality of groundwater samples using a Wilcox diagram.**

Wilcox diagram results indicate that the groundwater in the area is not suitable for drinking without treatment, as most of the samples fall in the high to very high salinity and SAR categories, and moderate conductivity and alkalinity categories (Fig 10). High SAR levels may negatively impact water quality, resulting in aesthetic and health issues, and long-term consumption of water with high salinity levels can have adverse health consequences. Scale accumulation in pipes and fixtures caused by high conductivity and alkalinity levels can cause infrastructure maintenance problems. To maintain the quality and reduce potential health risks, treating groundwater before consumption is essential.

**Table 2. Comparison of WQI and status evaluation using weighted arithmetic method: Tabulated summary of findings.**

| WQI | Rating Class | No. of samples | % of samples | % of area |
|---|---|---|---|---|
| 0–25 | Excellent | 9 | 22.5 | 4 |
| 26–50 | Good | 10 | 25 | 30 |
| 51–75 | Poor | 11 | 27.5 | 21 |
| 76–100 | Very poor | 1 | 2.5 | 10 |
| >100 | Not suitable | 9 | 22.5 | 35 |

## WQI

Using ArcGIS 10.5, a WQI map was created based on a selection of quality parameters to determine the quality classes of excellent, good, poor, very poor, and not suitable at each sampling station for drinking purposes. Tables 2 and 3 were used to prepare this map. The results of WQI map revealed that the majority of the study area has not suitable groundwater quality

**Table 3. Evaluation of hydro-stations water quality index and class in study area.**

| Sample No. | Sample source | WQI Value | Class |
|---|---|---|---|
| S1 | Hand pump | 57.71 | Poor |
| S2 | Hand pump | 22.25 | Excellent |
| S3 | Hand pump | 108.38 | Not suitable |
| S4 | Hand pump | 27.49 | Good |
| S5 | Hand pump | 107.02 | Not suitable |
| S6 | Hand pump | 68.15 | Poor |
| S7 | Hand pump | 30.10 | Good |
| S8 | Hand pump | 36.81 | Good |
| S9 | Hand pump | 70.61 | Poor |
| S10 | Hand pump | 214.93 | Not suitable |
| S11 | Hand pump | 221.48 | Not suitable |
| S12 | Hand pump | 238.02 | Not suitable |
| S13 | Tube well | 53.37 | Poor |
| S14 | Hand pump | 230.46 | Not suitable |
| S15 | Hand pump | 40.75 | Good |
| S16 | Hand pump | 24.47 | Excellent |
| S17 | Hand pump | 14.59 | Excellent |
| S18 | Hand pump | 32.80 | Good |
| S19 | Hand pump | 47.08 | Good |
| S20 | Hand pump | 92.14 | Very Poor |
| S21 | Hand pump | 206.17 | Not suitable |
| S22 | Hand pump | 19.59 | Excellent |
| S23 | Hand pump | 57.69 | Poor |
| S24 | Hand pump | 22.00 | Excellent |
| S25 | Hand pump | 33.62 | Good |
| S26 | Hand pump | 12.57 | Excellent |
| S27 | Hand pump | 75.13 | Poor |
| S28 | Hand pump | 74.04 | Poor |
| S29 | Hand pump | 40.94 | Good |
| S30 | Hand pump | 14.74 | Excellent |
| S31 | Hand pump | 50.88 | Poor |
| S32 | Hand pump | 75.42 | Poor |
| S33 | Hand pump | 20.66 | Excellent |
| S34 | Hand pump | 106.34 | Not suitable |
| S35 | Hand pump | 17.42 | Excellent |
| S36 | Hand pump | 60.80 | Poor |
| S37 | Boring well | 66.79 | Poor |
| S38 | Hand pump | 30.66 | Good |
| S39 | Hand pump | 32.18 | Good |
| S40 | Hand pump | 1014.07 | Not suitable |

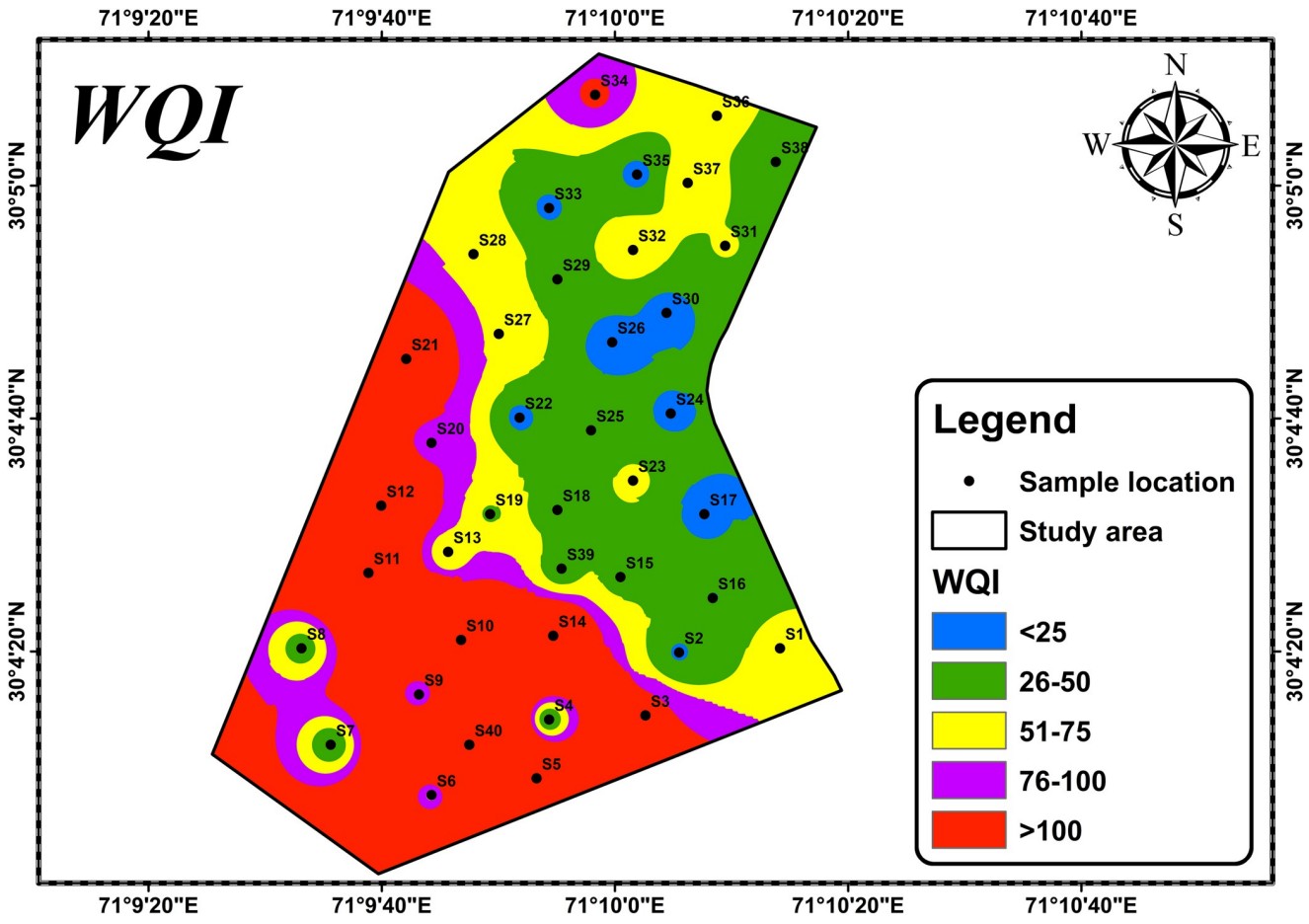

**Fig 11. Visualizing the spatial distribution of water quality using WQI scores.**

(>100), while only a few areas had excellent quality (0–25) (Fig 11; Table 2). The map also indicated that the groundwater quality in the north and northeast regions was excellent to good for human consumption. The outward direction from good quality groundwater showed a gradual decline in water quality from poor to very poor. The south-southwest region was characterized by not suitable to poor quality groundwater (Fig 11). Overall, the majority of the study area had not suitable to poor quality groundwater, making it not suitable for both drinking and domestic use. Table 2 presents the classification of water quality and status based on the weighted arithmetic WQI method, along with the percentage of the study area falling into each category.

## Discussions

The results of this study provide a comprehensive understanding of groundwater quality in the Thal desert region of Muzaffargarh, Pakistan, and its potential implications for human health. The geospatial analysis of water quality parameters has unveiled several concerning findings. Some groundwater samples exhibit undesirable color and odor, indicating potential contamination, possibly by organic matter, heavy metals, or microbial agents. Moreover, the presence of a salty or bitter taste in certain samples raises concerns about water suitability for

both drinking and agricultural purposes. Groundwater temperature and pH levels, which remained relatively consistent across the study area, hint at a stable geological and hydrogeological environment, with the water displaying a moderately alkaline nature due to $CO_2$ loss and mineral salt accumulation. High levels of TDS and EC in specific samples suggest significant mineralization, likely resulting from material weathering and human activities. Additionally, turbidity exceedance of WHO guidelines in some samples hints at the potential presence of suspended particles and pollutants.

The most alarming finding is the significantly elevated concentration of arsenic in the groundwater, well above WHO limits, posing severe health risks from prolonged exposure. High levels of water hardness may lead to scaling issues and affect water taste. Elevated levels of various ions and compounds such as sodium, potassium, magnesium, calcium, fluoride, chloride, nitrate, bicarbonate, and sulfate, along with microbial contamination, indicate a complex interplay of geological, hydrogeological, and anthropogenic factors affecting water quality. Ions such as calcium, magnesium, and bicarbonates are produced by geological processes such as mineral weathering. Mineral mobility under particular microbial-induced redox conditions results in elevated arsenic concentrations. Anthropogenic activities, notably excessive industrial processes, urbanization and waste disposal introduce nitrates, potassium, and chloride and further exacerbate groundwater pollution. Variations in sodium level are indicative of soil leaching, household pollution and industrial processes.

The hydrogeochemical evaluation of groundwater, illustrated through the Piper, Durov, Gibbs, and Stiff diagrams, provided insights into water composition and its origins. The dominance of specific cations and anions suggests the influence of weathering processes, ion exchange, and human activities. Correlation coefficients and a scatter plot matrix between various water quality parameters helped identify potential links, which are crucial for understanding contamination sources and processes affecting water quality. The composition of groundwater samples is visualized using pie charts, which reveal information on the dominant geochemical processes. According to the Schoeller diagram, the groundwater in the area is primarily homogenous, emanating from the same hydrogeological system, which has consequences for source monitoring and finding sources of contamination. With its high salinity and SAR levels, the Wilcox diagram made it clear that groundwater is Not suitable for human consumption without remediation. This emphasizes the significance of water treatment to ensure quality water for consumption and avoid health problems.

The analysis of groundwater samples from the study area reveals a cation concentration trend in the following order: $Na^+ > Ca^{2+} > Mg^{2+} > K^+$, highlighting sodium ($Na^+$) as the dominant ion. The presence of $Ca^{2+}$, $Na^+$, and $Mg^{2+}$ is typically linked to clay-forming minerals such as illite, montmorillonite, and chlorite. Calcium ions in groundwater primarily originate from the dissolution of minerals like calcium carbonate ($CaCO_3$) and dolomite ($CaMg(CO_3)_2$). Silicate weathering is a key physical process that introduces $Na^+$ ions into groundwater. During these reactions, calcium carbonate in soil reacts with carbonic acid, resulting in bicarbonate and calcium ions. Normalized scatter plots, exemplified by Fig 7, corroborate the processes of silicate and carbonate weathering in the study area. Silicate weathering significantly modifies the chemical composition of aquifer water, making it complex to assess silicates both quantitatively and qualitatively. The extent of silicate weathering can be gauged by examining the concentrations of $Na^+$ and $Ca^{2+}$ ions in the water samples. Minerals such as quartz, feldspar, pyroxene, amphibole, mica, and olivine are susceptible to silicate weathering, transforming into clays rich in $Na^+$ and $K^+$. The high sodium ion concentration found in the samples indicates prevalent silicate weathering in the aquifer materials, contributing significantly to the elevated sodium levels in groundwater. The reaction of carbonic acid with feldspar minerals in the presence of water releases bicarbonate ions ($HCO_3^-$), the dominant anion

in the aquifer water. In arid and semi-arid regions, the dissolution and deposition of minerals like calcite, dolomite, gypsum, and halite within aquifer water occur naturally. Groundwater salinity may stem from the dissolution of chloride-rich minerals, although the likelihood of halite dissolution in the research area is minimal due to sodium concentrations being higher than chloride levels. Ion exchange processes are crucial for understanding groundwater chemistry in arid and semi-arid regions, particularly in recharge areas with high cation exchange capacities and sodium present on exchange sites. Recognizing the various hydrogeochemical processes that groundwater undergoes as it moves through different subsurface formations is vital. Bivariate graphs further support evidence of the cation exchange process in this study area. Overall, this study provides a comprehensive understanding of the hydrogeochemical processes influencing groundwater quality in the southern Thal Desert, emphasizing the dominant role of silicate weathering and ion exchange in shaping the groundwater's chemical composition.

Ultimately, the WQI map revealed that a significant proportion of the groundwater in the research region is unfit for human consumption, highlighting the critical need for improved management, monitoring, and remediation of the water quality in order to protect the health and welfare of the residents of the area. In order to ensure that the Thal desert region has a supply of clean and wholesome water to consume, it is imperative that these water quality issues are resolved. A study was conducted in the southern end of Thal Doab, where it borders Chenab River [2]. The region is delineated by Muzaffargarh town to the south and the settlements of Rangpur in the northeast and Chowk Sarwar Shaheed in the north. The study area encompasses abandoned floodplain terraces of the Chenab River, overlaid by windblown sands from the Thal Desert at inland sites (from site 14 northwards to Chowk Sarwar Shaheed and from site 33 westward to the same location). The study focused a comprehensive analysis of groundwater chemistry, emphasizing arsenic distribution and its unexpected behavior in the Muzaffargarh region. The authors skillfully explored various potential sources, particularly highlighting the role of local reductive dissolution of iron oxides driven by organic pollution. The findings contributed valuable insights into the complex interplay of geological and anthropogenic factors influencing groundwater quality. Moreover, the study adeptly addressed the impact of irrigation on major constituents and successfully integrates relevant geochemical indicators. Overall, the paper demonstrated a thorough investigation and thoughtful interpretation of groundwater dynamics in the studied area.

Both studies on Muzaffargarh's groundwater exhibit notable consistencies and advancements. Both recognize arsenic contamination, major ion concentration trends, and the impact of natural and anthropogenic factors. However, the recent study surpasses its predecessor by delving deeper into the effects of irrigation, unveiling specific mechanisms like local reductive dissolution, and employing a more comprehensive range of analytical tools. The refined exploration of spatial patterns and the nuanced focus on urban areas as arsenic contributors distinguish the latest research. The new study, thus, not only reaffirms the foundational findings of the earlier work but also presents a more detailed and contemporary understanding of the complex hydrogeochemical processes in Muzaffargarh.

## Conclusions

The Thal Desert is composed of Quaternary alluvial deposits that originated from the Himalayan rocks through the Indus River. This study aimed to determine the physicochemical parameters, like major ions, and trace elements in groundwater samples from Muzaffargarh city, a region of the Thal Desert. The groundwater in this area is heavily salinized and contaminated with microorganisms, affecting a large portion of the population that relies on it. The analysis

of 40 semi-deep wellbore samples revealed that the aquifer chemistry is influenced by surficial alluvial deposits and geogenic and anthropogenic inputs. The cationic variation of the samples was $Na^+ > Ca^{2+} > As > Mg^{2+} > K^+ > Fe^{2+}$, while the anionic concentrations were $Cl^- > SO_4^{2-} > HCO_3^- > F^- > NO_3^-$. The presence of $Ca^{2+}$, $Na^+$, $Mg^{2+}$, $K^+$, $HCO_3^-$, and F- is due to geogenic inputs, while $Cl^-$, $SO_4^{2-}$, and $NO_3^-$ are due to anthropogenic impacts. These findings can help improve the management of groundwater resources in the region. The Piper diagram categorizes groundwater into six distinct types based on its chemical composition. The majority of groundwater samples were found to be bicarbonate, sodium, and chloride ions, which are the dominant ionic species due to weathering processes and ion exchange. The Durov diagram reveals that the mixed type was the most common groundwater type, and significant variation was observed in the anion composition of groundwater samples. The Gibbs diagram shows the factors that influence groundwater chemistry, including precipitation, rock weathering, and evaporation. Mineral groundwater enrichment was primarily attributed to rock weathering, and anthropogenic activities such as domestic and industrial effluents can also affect groundwater chemistry. The Stiff plots identify the dominant geochemical processes affecting groundwater quality, and the scatter plot matrix visually displays the relationships between multiple physicochemical parameters. Pie charts compare the composition of different samples to identify dominant parameters affecting water quality. The Schoeller diagram reveals a consistent trend in the curve of water samples, indicating that the groundwater sources in the region are relatively homogeneous. According to the findings from the Wilcox diagram, it is evident that untreated groundwater in the studied area is Not suitable for drinking. Additionally, ArcGIS 10.5 software was utilized to generate a water quality index map. The results of the map indicate that a significant portion of the study area exhibits poor quality groundwater, with only a few localized areas demonstrating good to excellent quality.

## Recommendations

The current investigation indicates a deterioration in groundwater quality attributed to widespread salinity, significant fecal, arsenic, and fluoride pollution. Consequently, numerous individuals in the study area are grappling with bacterial, arsenic, and fluoride-induced ailments. It is imperative to conduct comprehensive geophysical and physiochemical studies to assess the suitability of surface and groundwater for drinking and domestic purposes in both the study area and other regions of Muzaffargarh city. These studies aim to identify pollution levels, types, and sources (geogenic and anthropogenic).

Several recommendations emerge from this study:

- The municipal water system should implement filtration and chlorination processes for available surface water to mitigate waterborne diseases such as gastroenteritis, cholera, and diarrhea.

- Arsenic field kit tests should be employed to screen wells in the area. Arsenic-affected wells should be marked in red, while arsenic-free wells should be identified with green markings.

- In regions significantly impacted by salinity and arsenic contamination, the installation of new hand pump wells should be based on identifying fresh water and arsenic-free aquifers.

- Discouraging the use of pit latrine systems is advised due to the increased risk of microbial, fluoride, and arsenic contamination.

- Improving sanitation facilities and establishing a proper sewerage line system for waste disposal in affected areas is crucial.

- Conduct microbial testing of all available surface and groundwater sources in the area.

- Implement cost-effective disinfection units, such as UV filters, to address microbiological contamination in affected areas.

- Encourage the installation of deeper hand pumps to prevent contamination from surface runoff.

- Conduct a detailed medical examination of the local population to identify the occurrence and severity of various diseases.

- Develop a small-scale map based on detailed studies in other parts of Muzaffargarh city regarding the drinking quality of groundwater.

By implementing these recommendations, it is anticipated that the overall water quality in the study area and Muzaffargarh city can be significantly improved, thereby safeguarding the health and well-being of the local population.

## Supporting information

**S1 File. Physiochemical data of groundwater samples.**
(XLSX)

**S2 File. Data for Piper, Durov, and Stiff plots.**
(XLSX)

**S3 File. Data for Gibbs diagram.**
(XLSX)

**S4 File. Data for Schoeller plot.**
(XLSX)

**S5 File. Data for Wilcox diagram.**
(XLSX)

**S6 File. Calculation of WQI.**
(XLSX)

## Author Contributions

**Conceptualization:** Irfan Raza.

**Data curation:** Irfan Raza, Perveiz Khalid, Qazi Adnan Ahmad.

**Formal analysis:** Irfan Raza, Qazi Adnan Ahmad, Shahzada Khurram.

**Investigation:** Irfan Raza.

**Methodology:** Irfan Raza, Rabia Zainab.

**Project administration:** Irfan Raza.

**Resources:** Irfan Raza, Shahzada Khurram, Salman Farooq.

**Software:** Irfan Raza.

**Supervision:** Perveiz Khalid, Muhammad Irfan Ehsan.

**Validation:** Irfan Raza, Muhammad Irfan Ehsan, Rabia Zainab.

**Visualization:** Irfan Raza, Salman Farooq.

**Writing – original draft:** Irfan Raza.

**Writing – review & editing:** Irfan Raza, Perveiz Khalid.

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
