## [Decision Letter · Decision Letter 0]

13 Dec 2023

PONE-D-23-39260Geospatial Interpolation and Hydro-geochemical Characterization of Physicochemical Parameters in Unconfined Alluvial Aquifers of the Thal Desert, Punjab, PakistanPLOS ONE

Dear Dr. Ehsan,

Thank you for submitting your manuscript to PLOS ONE. After careful consideration, we feel that it has merit but does not fully meet PLOS ONE’s publication criteria as it currently stands. Therefore, we invite you to submit a revised version of the manuscript that addresses the points raised during the review process.

We look forward to receiving your revised manuscript.

Kind regards,

Venkatramanan Senapathi, Ph.D.

Academic Editor

PLOS ONE

Journal Requirements:

4. We note that Figures 1,2 and 11 in your submission contain map/satellite images which may be copyrighted. All PLOS content is published under the Creative Commons Attribution License (CC BY 4.0), which means that the manuscript, images, and Supporting Information files will be freely available online, and any third party is permitted to access, download, copy, distribute, and use these materials in any way, even commercially, with proper attribution. For these reasons, we cannot publish previously copyrighted maps or satellite images created using proprietary data, such as Google software (Google Maps, Street View, and Earth). For more information, see our copyright guidelines: http://journals.plos.org/plosone/s/licenses-and-copyright.

a. You may seek permission from the original copyright holder of Figures 1,2 and 11 to publish the content specifically under the CC BY 4.0 license.  

Additional Editor Comments:

The study on the hydrogeochemical characteristics of groundwater in the southern part of Thal Desert of Pakistan appears to be thorough in its analysis, but its findings are disheartening. The extensive contamination sources, including mineral infiltration, domestic waste, and industrial effluents, paint a grim picture of the groundwater quality. The prevalence of salinity and the identified factors contributing to elevated levels of various ions indicate a concerning degradation of the water. The utilization of visual aids and indices may add a layer of complexity, but the overall revelation that 52.5% of the samples exhibit poor to unsuitable quality is alarming. The study underscores significant challenges in maintaining water quality in the region, leaving little room for optimism.

Reviewers' comments:

Reviewer's Responses to Questions

**Comments to the Author**

1. Is the manuscript technically sound, and do the data support the conclusions?

Reviewer #1: Partly

Reviewer #2: Yes

2. Has the statistical analysis been performed appropriately and rigorously? 

Reviewer #1: No

Reviewer #2: Yes

3. Have the authors made all data underlying the findings in their manuscript fully available?

Reviewer #1: Yes

Reviewer #2: Yes

4. Is the manuscript presented in an intelligible fashion and written in standard English?

Reviewer #1: No

Reviewer #2: No

5. Review Comments to the Author

Reviewer #1: To,

Editor

Journal of PLOS ONE

Subject:-Geospatial Interpolation and Hydro-geochemical Characterization of Physicochemical Parameters in Unconfined Alluvial Aquifers of the Thal Desert, Punjab, Pakistan, PONE-D-23-39260

Dear Sir,

I am submitting the following for the consideration,

1.Title may reduced as if possible.

2. L226 suspects high amount of turbidity in groundwater, mostly it may be within limits.

3. The complete physical chemical parameters of each sample, please indicated in separate table.

4. L141 incubation temperature may conformed, coliforms, ecoli were not discussed.

5. The results and discussion that did not referenced and studies may find correlation with global or local studies.

6. The figures are of poor resolution 300dpi.

7. The temporal and spatial an objective, that may explained well with the references.

8. The samples may classified as geogenic and anthropogenic.

Reviewer #2: I have following observations and suggestions

Hydrogeology of the area should be provided in detail

How the analytical quality maintained and validated

The discussion part is very poor, need to discuss results in detail with causes and effects. It is not clear from where the contaminants are added to groundwater

There are very few cited papers on the hydrogeology of Indus basin and groundwater status of the area, consult following articles and cite

MacAllister, et al 2022. A century of groundwater accumulation in Pakistan and northwest India. Nature Geoscience. https://doi.org/10.1038/s41561-022-00926-1

Bonsor et al 2017. Hydrogeological typologies of the Indo-Gangetic basin alluvial aquifer, South Asia. Hydrogeology Journal. 25(5): 1377-1406 DOI: 10.1007/s10040-017-1550-z

6. PLOS authors have the option to publish the peer review history of their article (what does this mean?). If published, this will include your full peer review and any attached files.

Reviewer #1: No

Reviewer #2: No

---

## [Author Response · Author response to Decision Letter 0]

24 Jan 2024

The authors are very thankful to the editor and the reviewers for their valuable time and comments. We have revised the manuscript in the light of the suggestions and comments of the reviewers and the editor. Please see the “Revised Manuscript (Marked up copy).docx”. All important changes made in the manuscript are tracked using the 'Track Changes' in the manuscript file.

The response of each comment is given below in tabular form. We are hopeful that the manuscript is now ready for acceptance.

Kind regards,

Muhammad Irfan Ehsan (Ph.D.)

---

## [Decision Letter · Decision Letter 1]

13 Feb 2024

PONE-D-23-39260R1Geospatial interpolation and hydro-geochemical characterization of alluvial aquifers in the Thal Desert, Punjab, PakistanPLOS ONE

Dear Dr. Ehsan,

Thank you for submitting your manuscript to PLOS ONE. After careful consideration, we feel that it has merit but does not fully meet PLOS ONE’s publication criteria as it currently stands. Therefore, we invite you to submit a revised version of the manuscript that addresses the points raised during the review process.

We look forward to receiving your revised manuscript.

Kind regards,

Venkatramanan Senapathi, Ph.D.

Academic Editor

PLOS ONE

Reviewers' comments:

Reviewer's Responses to Questions

**Comments to the Author**

1. If the authors have adequately addressed your comments raised in a previous round of review and you feel that this manuscript is now acceptable for publication, you may indicate that here to bypass the “Comments to the Author” section, enter your conflict of interest statement in the “Confidential to Editor” section, and submit your "Accept" recommendation.

Reviewer #1: (No Response)

Reviewer #2: All comments have been addressed

2. Is the manuscript technically sound, and do the data support the conclusions?

Reviewer #1: No

Reviewer #2: Yes

3. Has the statistical analysis been performed appropriately and rigorously? 

Reviewer #1: No

Reviewer #2: Yes

4. Have the authors made all data underlying the findings in their manuscript fully available?

Reviewer #1: No

Reviewer #2: No

5. Is the manuscript presented in an intelligible fashion and written in standard English?

Reviewer #1: No

Reviewer #2: Yes

6. Review Comments to the Author

Reviewer #1: To,

Editor

Journal of PLOS ONE

Subject:-Geospatial interpolation and hydro-geochemical characterization of alluvial aquifers in the Thal Desert, Punjab, Pakistan. PONE-D-23-39260R1

Dear Sir,

I am submitting the following for the consideration;

1. The manuscript did not explained well and short.

2. The table of physicochemical parameters of all 40 samples may also provided with sampling names and GIS locations.

3. Introduction was short with less relevant information.

4. The samples may explained with respect to spatial variation and hierarchal analysis.

5. The microbial count may explained in groundwater as indicated Line 284, this may added as separate section.

6. The water dependency of total population and houses on each sample may also explained and may present in table.

7. The ion balance error may also explained.

8. Point 2 not acceptable as high turbidity of most handpumps.

9. Point 5 did not responded by addition of references of relevant studies from local or abroad. Although literature present for water quality of Muzaffargarh that did not included here.

10. Point 6 the resolution may improve if possible.

11. Kindly explain the purpose of using following diagrams Durov, and Gibbs diagrams, scatter plots, stiff and Schoeller.

12. Please correct in conclusion, the arsenic remains lower mostly; but here it indicated higher than magnesium and potassium.

13. The results were not compared with the local studies or international.,

14. It may also added the literature, similar from deserts for the present data for its comparisons.

15. The figures presented that were not clear, particularly figure 2.

16. SAR for agriculture may also explained.

17. The experimental may explained well i.e microbiol analysis.

18. The samples may explained on basis with cluster depending on depth, direction, desert or non desert etc, principal analysis may also applicable.

19. The Pearson correlation did not explained.

Reviewer #2: (No Response)

7. PLOS authors have the option to publish the peer review history of their article (what does this mean?). If published, this will include your full peer review and any attached files.

Reviewer #1: **Yes: **Taj Muhammad Jahangir

Reviewer #2: No

---

## [Author Response · Author response to Decision Letter 1]

16 Mar 2024

The authors are very thankful to the editor and the reviewers for their valuable time and comments. We have revised the manuscript in the light of the suggestions and comments of the reviewers and the editor. Please see the “Revised Manuscript (Marked up copy).docx”. All important changes made in the manuscript are tracked using the 'Track Changes' in the manuscript file.

The response of each comment is given in tabular form. We are hopeful that the manuscript is now ready for acceptance.

Kind regards,

Muhammad Irfan Ehsan (Ph.D.)

---

## [Decision Letter · Decision Letter 2]

29 Apr 2024

PONE-D-23-39260R2Geospatial interpolation and hydro-geochemical characterization of alluvial aquifers in the Thal Desert, Punjab, PakistanPLOS ONE

Dear Dr. Ehsan,

Thank you for submitting your manuscript to PLOS ONE. After careful consideration, we feel that it has merit but does not fully meet PLOS ONE’s publication criteria as it currently stands. Therefore, we invite you to submit a revised version of the manuscript that addresses the points raised during the review process.

**This study on groundwater hydrogeochemistry in the southern Thal Desert of Pakistan offers valuable insights, yet it could benefit from minor improvements. While the study effectively aims to identify contamination sources and assess their impact on groundwater and the ecosystem, there's room for clearer quantification of pollutant levels from identified sources. Additionally, the explanation of weathering processes and ion exchange effects could be more detailed. Providing more context for the results within existing literature would enhance the study's significance. Furthermore, the relatively small sample size might limit the generalizability of the findings, and the focus solely on certain groundwater sources may overlook other potential contributors to contamination. Lastly, while the study categorizes water quality, it lacks detailed justification for the chosen thresholds. Addressing these minor issues would strengthen the study's overall impact and relevance.**

We look forward to receiving your revised manuscript.

Kind regards,

Venkatramanan Senapathi, Ph.D.

Academic Editor

PLOS ONE

Additional Editor Comments:

This study on groundwater hydrogeochemistry in the southern Thal Desert of Pakistan offers valuable insights, yet it could benefit from minor improvements. While the study effectively aims to identify contamination sources and assess their impact on groundwater and the ecosystem, there's room for clearer quantification of pollutant levels from identified sources. Additionally, the explanation of weathering processes and ion exchange effects could be more detailed. Providing more context for the results within existing literature would enhance the study's significance. Furthermore, the relatively small sample size might limit the generalizability of the findings, and the focus solely on certain groundwater sources may overlook other potential contributors to contamination. Lastly, while the study categorizes water quality, it lacks detailed justification for the chosen thresholds. Addressing these minor issues would strengthen the study's overall impact and relevance.

Reviewers' comments:

Reviewer's Responses to Questions

**Comments to the Author**

1. If the authors have adequately addressed your comments raised in a previous round of review and you feel that this manuscript is now acceptable for publication, you may indicate that here to bypass the “Comments to the Author” section, enter your conflict of interest statement in the “Confidential to Editor” section, and submit your "Accept" recommendation.

Reviewer #1: (No Response)

2. Is the manuscript technically sound, and do the data support the conclusions?

Reviewer #1: No

3. Has the statistical analysis been performed appropriately and rigorously? 

Reviewer #1: No

4. Have the authors made all data underlying the findings in their manuscript fully available?

Reviewer #1: No

5. Is the manuscript presented in an intelligible fashion and written in standard English?

Reviewer #1: No

6. Review Comments to the Author

**Reviewer #1:** To,

Editor

PLOS ONE

Subject:- Geospatial interpolation and hydro-geochemical characterization of alluvial aquifers in the Thal Desert, Punjab, Pakistan. PONE-D-23-39260R2

Dear Sir,

I submit the following my comments a review for the consideration.

1. The manuscript is still short with less discussions and needs improvements and corrections throughout manuscript.

2. The TDS may derived from electrical Conductivity by multiply the 0.64 on standard temperature.

3. Abnormally in many cases the Mg almost equal to Ca.

4. The turbidity as WHO guidelines not acceptable >5 NTU for waters. The closed unhindered systems lake in groundwater the turbidity are questionable, standard methods and sampling are highly desirable for the publications.

5. The microbiology in groundwater may questionable and explained the reasons.

6. How the TDS upto 500mg/l indicated the taste of water unpleasant.

7. The points 1 and 4 as the spatial variations that did not well explained.

8. The points 6,7, 8, 9, 13, 14, 18 did not well explained.

9. The figures did not yet improved.

10. The results discussion were not compared with local and the world studies.

11. The general order of As (as this was ppb level) that did not corrected in conclusion.

12. The recommendations may reduced with importance of present work.

13. The samples may discussed with each other, rather in groups.

7. PLOS authors have the option to publish the peer review history of their article (what does this mean?). If published, this will include your full peer review and any attached files.

Reviewer #1: No

---

## [Author Response · Author response to Decision Letter 2]

24 Jun 2024

Thank you for your constructive feedback on our manuscript. We have addressed several key points in the revision process to enhance the clarity and significance of our study. Specifically, we have provided a detailed explanation of weathering processes and ion exchange effects in the discussion section, spanning from lines 485 to 509. This addition aims to provide a comprehensive understanding of these factors as they relate to groundwater hydrogeochemistry in the southern Thal Desert of Pakistan.

Regarding the quantification of pollutant levels, we have improved our analysis to better quantify pollutant levels from identified sources. Additionally, we have provided more context for our results within the existing literature to enhance the study's significance.

We acknowledge the limitation of the relatively small sample size and have provided a clearer discussion on the generalizability of our findings.

We believe these revisions strengthen the manuscript and address the minor issues raised.

---

## [Editor Report · Decision Letter 3]

28 Jun 2024

Geospatial interpolation and hydro-geochemical characterization of alluvial aquifers in the Thal Desert, Punjab, Pakistan

PONE-D-23-39260R3

Dear Dr. Ehsan,

We’re pleased to inform you that your manuscript has been judged scientifically suitable for publication and will be formally accepted for publication once it meets all outstanding technical requirements.

Kind regards,

Venkatramanan Senapathi, Ph.D.

Academic Editor

PLOS ONE

Additional Editor Comments (optional):

Based on the reviewer's comments, the authors have thoroughly revised the manuscript. The revisions address the concerns and suggestions provided, enhancing the reliability With these improvements, the manuscript is now suitable for publication.